# Addressing Problems beyond Heritage, Patrimony, and Representation: Reflections on Twenty Years of Community Archaeology in the Southwestern Maya Lowlands

**Brent K.S. Woodfill** [1,*] and **Alexander E. Rivas** [2]

1  Department of Sociology, Criminology, and Anthropology, Winthrop University, Rock Hill, SC 29733, USA
2  Department of Anthropology, Washington University in St. Louis, St. Louis, MO 63130, USA; arivas@wustl.edu
*  Correspondence: woodfillb@winthrop.edu

**Abstract:** Collaborative or community archaeology as a methodological approach has a long history and is becoming increasingly common in the Maya world. This article draws from the authors' experiences on three distinct archaeological projects to discuss the benefits and obstacles we confronted while conducting collaborative research with contemporary Maya communities as well as lessons we learned that can increase the odds of a mutually beneficial partnership. After summarizing the history of the research projects and the expectations for and contributions of the scientific and community stakeholders, we propose several characteristics that were particularly helpful. These include the need for all parties to engage in sincere and sustained dialogue, to be flexible, and to take others in account when making any plans that affect them. Most importantly, we urge archaeologists to collaborate with community endeavors beyond those that are directly related to their research, offering a few examples of how archaeological skills, equipment, and social capital can be used to address a wide range of local concerns beyond patrimony and heritage.

**Keywords:** community archaeology; Maya archaeology; community development; archaeological ethics; world heritage; continuity

## 1. Introduction

In recent years, an increasing number of archeologists have embraced the inherently political nature of our field and worked to share some tangible advantages with the communities most affected by our presence and interpretations. In such community-oriented programs, the design, implementation, and results of archaeological investigation are undertaken in consultation with local populations. These communities are, to varying degrees, involved with the project design, are an active voice in data collection, analyses, and interpretation, and can use the research to further their own community initiatives.

In this manner, archaeologists do not simply work with or near people, but work "*for living communities*" ([1], emphasis ours) as we strive to create a truly anthropological archaeology [2]. Often, archaeologists use this perspective for heritage initiatives including archaeological tourism, site consolidation, and training locals as tour guides. This is a common and successful approach in the Maya region, with many archaeological projects successfully adding tourism, heritage protection, and community development initiatives towards their research goals [3–9].

However, at its core, Western science is an extractive industry, one has been intertwined with colonialism at least since the publication of *Leviathan* in 1651. In this text, Thomas Hobbes argued that his European ancestors forged a social contract in which the majority ceded power to the best

and brightest among them, creating the institution of kingship and, along with it, a superior society that was justified in conquering the world. Subsequent scientists—including our own founding figure Franz Boas—disinterred and dismembered Indigenous bodies, looted Indigenous tombs and temples, and exploited Indigenous knowledge in the name of objectivity, truth, and the advancement of knowledge [10,11] (pp. 181). In light of this troubled past, we feel that archaeology and other social sciences have a moral imperative to engage explicitly in decolonizing methodologies, especially in settings where Indigenous and other marginalized communities are actively affected by the research. Best stated by Linda Tuhiwai Smith, decolonization efforts are about "centering [Indigenous] concerns and worldviews and then coming to know and understand theory and research from [their] perspectives, and for [their] own purposes [12], p. 41".

When engaging with Indigenous communities, community-oriented archaeology can and should take a decolonialist approach that is sincerely engaged with the worldviews and concerns of the communities where we work. This approach stands in stark contrast to the fundamental scientific paradigm, in which Science is undertaken for the "good of mankind", even if little to none of said good trickles down to the communities affected and inconvenienced by our research [12–14]. In the archaeological context, decolonizing methodologies have typically meant privileging the knowledge, memories, and spiritual aspects of archaeological sites and materials [3,4,15–21]. With this focus, archaeologists support Indigenous groups as they fight for the stewardship and protection of their own cultural heritage and resources, with archaeologists taking a reflexive approach and acting as collaborators rather than managers, owners, or even the primary experts of cultural heritage [22].

However, in the southwestern Maya lowlands, concern for the protection, interpretation, and ownership of ancient Maya cultural resources is often not the primary concern of the economically and politically marginalized Indigenous communities, many of which lack access to a reliable source of clean water, medicine, land titles, and sustainable income. While we agree that foreign archaeologists can and should support Indigenous archaeologies and fight alongside descendant communities to wrest their heritage back from the heirs of colonial powers, we believe that archaeologists better serve these communities by being a transient (for even multi-year investigations must end) toolkit to address issues and problems of *their* choosing. The skills, technologies, and connections archaeologists can provide—equipment for survey and excavation, scientific knowledge, experience writing successful grants and presenting to different publics, and ties to the press and groups and organizations at nearly every rung of society (farmers, local governmental offices, NGOs, etc) are often much more valuable for locals than the actual act of conducting archaeology and the paradigms and products that result from it.

Our approach as researchers is based on this desire to empower local initiatives rather than impose our own values upon them, and can be described as both collaborative and community-based. While community members are actively involved as stakeholders in nearly all aspects of research, we make space for our presence to be used to address concerns, problems, and goals identified by the community members themselves [22,23].

Both of the present authors have been engaged in community archaeology for multiple years with the overarching goal of breaking out of the extractive research paradigm that has been the norm for much of the history of our field and which largely limits the benefits of our investigations to members of the academy. In this article, we draw on our experiences conducting community-based research and collaborative practices that are advantageous to both local initiatives and foreign scientists alike in the southwestern Maya lowlands. The article that follows is written from our own perspective as non-Indigenous North Americans, albeit ones who have lived in and worked closely with contemporary Q'eqchi' Maya communities for much of our professional careers.

## 2. The History of Archaeology and Community Relations in the Southwestern Maya Lowlands

The Precolumbian residents of the southwestern Maya lowlands in present-day Guatemala and Mexico took advantage of their unique geology and strategic location to transform their polities into

economic powerhouses. Located at the base of the highlands near the headwaters of several major rivers, local residents of cities, towns, and hamlets alike were integral to the economy of the entire Maya world. The city of Salinas de los Nueve Cerros surrounded the only lowland non-coastal salt source, which its residents exploited by producing up to 24,000 metric tons of salt per year throughout its more than 2000 year history [24–26]. The longest river in in Mesoamerica (albeit divided into differently-named segments—the Negro, Chixoy, Salinas, and Usumacinta) cuts through the city, not only facilitating transportation of commodities but also providing large quantities of fish for salting and deep, fertile layers of volcanic soil for large-scale agriculture. To its east, the city of Cancuén was located at the headwaters of the Pasión River, allowing its residents to become the transportation and production hub for jade and other sumptuary goods during its brief florescence in the seventh and eighth centuries A.D. [27,28]. Smaller sites along the land and river routes serviced the merchants, travelers, and pilgrims who visited the myriad caves in the region in order to petition safe passage from the regional earth deities [26,29].

When the Spaniards arrived in the early sixteenth century, the region was still home to multiple Ch'ol Maya kingdoms. The Ch'ol were the linguistic and cultural heirs to the great cities of the southern lowlands, and they were still involved in the production and exchange of multiple important commodities including salt, cacao, and achiote [26,30,31]. Unlike the Maya of the Guatemalan highlands, Chiapas, and the Yucatan Peninsula, these kingdoms were able to escape colonization and incorporation into the Spanish Empire for nearly 200 years, only succumbing to a major offensive in the 1690s [32]. Although the Spaniards initially were content to christen and occupy the Indigenous settlements, by the early eighteenth century every Maya man, woman, and child they could find was rounded up and sent to areas firmly under Spanish control, leaving the southwestern lowlands virtually unoccupied.

The region only began to be repopulated to a significant degree in the 20th century through several waves of colonization, beginning with Q'eqchi' Maya serfs through the mid-1940s who were escaping the harsh living conditions of German coffee plantations in their ancestral homeland around the highland city of Coban [33]. Beginning in the 1950s, the Guatemalan government sponsored several programs encouraging the landless poor to move into the region and cut back the wilderness that had grown up in the three centuries of abandonment. As a result, the area became a multilingual mosaic composed of small villages of highland Q'eqchi' and K'iche' Maya, as well as native Spanish speakers from the Pacific coast. After the revolution of 1954, much of the land was given to the political elite associated with a succession of military dictatorships [26,34–36].

As the civil war began to heat up in the late 1970s, the southwestern lowlands became one of the most dangerous places in Guatemala to be Indigenous. One hundred and sixty one massacres occurred in and around the region between 1978 and the signing of the Peace Accords in 1996 [37], pp. 224–244, and the combination of overt violence, intimidation, and corruption at all levels of the government forced the local Maya to organize and fend for themselves to preserve the few resources they had access to [26,34–36]. While the end of the civil war reduced the threat of annihilation, subsequent years introduced new threats in the form of transnational corporations, drug cartels, and environmental NGOs pushing for the forced removal of communities in the name of nature conservation [26,36].

## 2.1. Community Archaeology in the Southwestern Maya Lowlands

Although archaeologists have long noted the importance of this region for understanding ancient economics and politics ([38–40]), the difficult political situation described above discouraged all but a few small archaeological projects [21,41–46], most of which were conducted before the flood of new residents reached their field sites. As a result, the community archaeology initiatives described in this article are the first to be conducted here.

The senior author of this article has engaged in three community archaeology projects since 2000, first as a graduate student on Vanderbilt University's Cancuén Archaeological Project (2000–2007, Guatemala), then as the director of Proyecto Salinas de los Nueve Cerros (2009-present, Guatemala) and

a series of exploratory projects over the border in eastern Chiapas, Mexico (2017-present). The junior author has participated in the last two projects to varying degrees since 2015. Each of these projects (Figure 1) shares a multi-site focus and a spirit of collaboration among multiple stakeholders that includes students and professionals from multiple scientific disciplines, neighboring communities, landowners, and aid organizations.

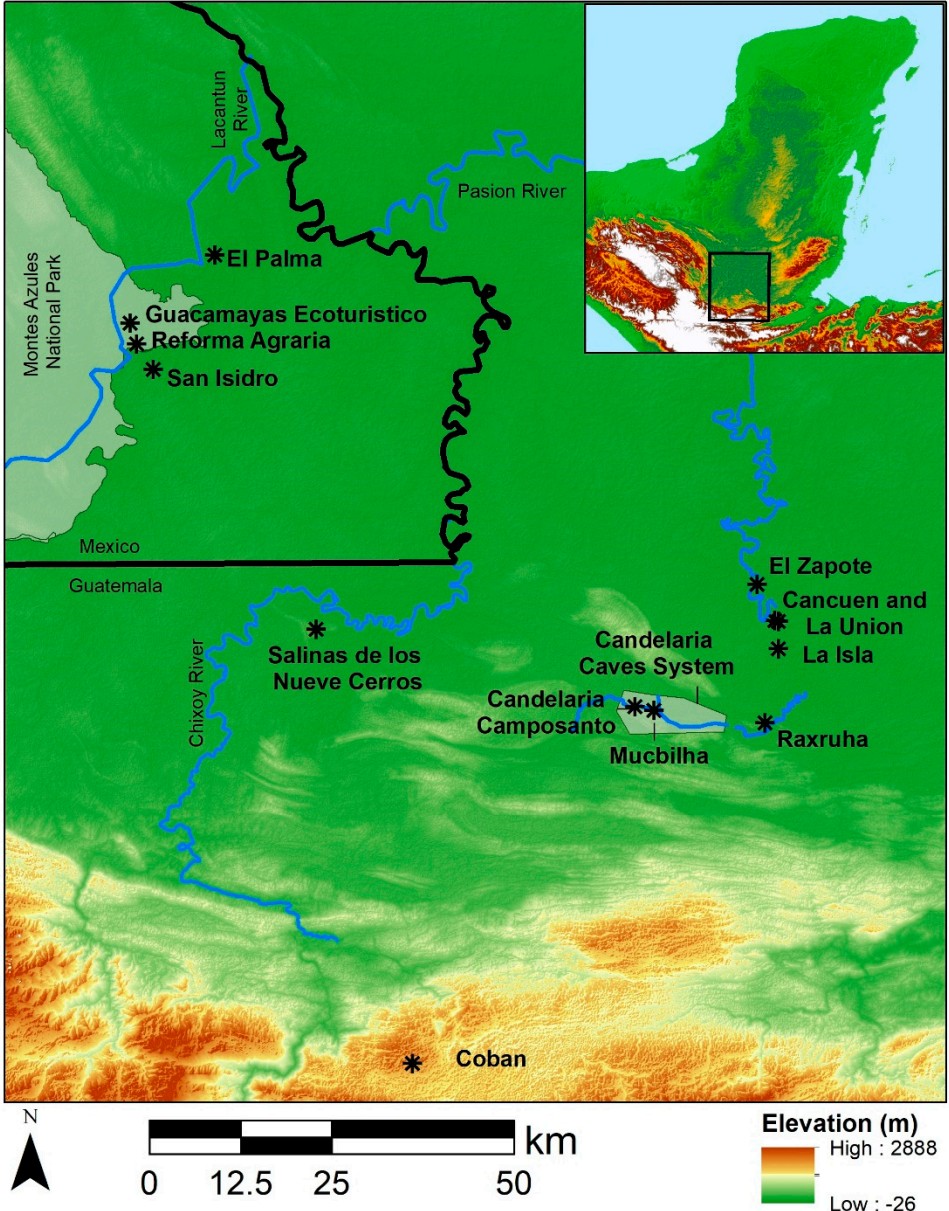

**Figure 1.** Archaeological sites, communities, and geographic features mentioned in the article. Map by A.E. Rivas.

## 2.2. The Cancuén Archaeological Project

The Cancuén Archaeological Project, led by Arthur A. Demarest (Vanderbilt University), has been intertwined with local Maya communities since its inaugural year. Project members arrived in the region in 1998 in order to investigate the nature of highland-lowland interaction in the years leading up to the Classic collapse (ca. A.D. 680–900) and rented a house in a village upriver from the site. They commuted down the Pasión River to the field by boat and employed residents from a second village across the river from the site as fieldworkers. While the team planned their research

program using established archaeological protocol—obtaining permission from the federal government, hiring locals as day laborers, and returning to an urban laboratory with the season's materials after closing excavations—they were immediately confronted with political problems. Even though they were working in an ostensibly uninhabited archaeological park with local villagers, unbeknownst to them, a third village that neighbored the site to the north claimed rights over the land. Due to tensions dating back to the Guatemalan civil war, there existed a high degree of unresolved tension among all three communities, and the archaeologists inadvertently reopened and exacerbated the conflict. As a result, the team found itself in the middle—literally and figuratively—of multiple armed standoffs.

Demarest used community archaeology, community development, community engagement, and the creation and sponsoring of a soccer league as the keys to mitigating this conflict, which over the next several years evolved into a collaborative juggernaut that necessitated the creation of multiple local NGOs and an entire subproject composed of ethnographers, ecotourism specialists, and development personnel [4,9,26]. As the local communities became more engaged and invested in the research, the project expanded into other parts of the region, and currently covers an area of over 1500 km$^2$ along a stretch of the highland-lowland transition in central Guatemala.

### 2.3. The Cancuén Cave Subproject

After a site visit the previous year, Woodfill joined the project in 2001, focusing on one primary research question—since Cancuén was a major city devoid of pyramids and most other standard ritual architecture, where were its residents performing rituals? Demarest suggested that the pyramidal karst hills riddled with caves that dotted the area (Figure 2) were the most likely suspects based on the ubiquity of cave worship in Mesoamerica [47,48] and epigrapher David Stuart's [49] observation that pyramids were referred to as "mountains" in Classic hieroglyphic texts. Woodfill set out to find data to test this hypothesis.

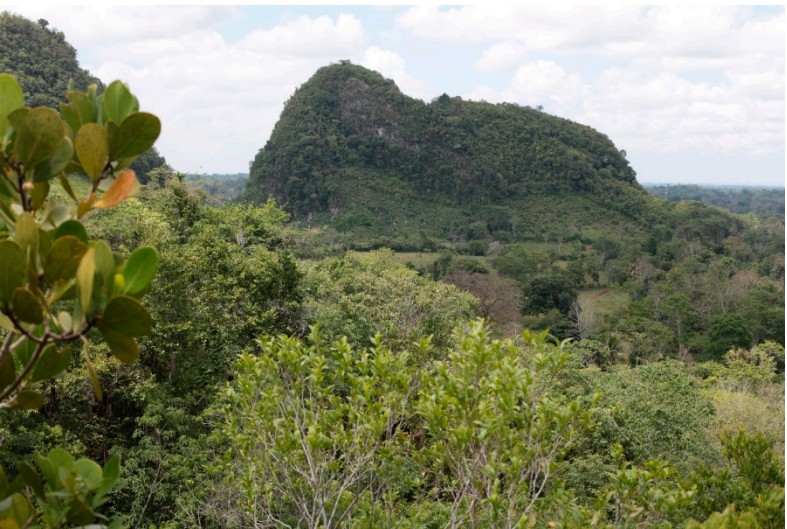

**Figure 2.** Haystack karst hills surrounding Cancuén. Photo by B.K.S. Woodfill.

He quickly discovered three primary logistic problems that complicated the quest for data. Each of these hill-cave zones was too far from camp to commute, all of them were sacred places where regular rituals were performed by the contemporary Q'eqchi' who moved into the area within the last 80 years, and access to them was only possible by passing through land that was divided among the myriad villages in the area. His strategy to overcome all three problems was simple: he would arrive in a village, meet with the community leaders and explain his research project, his needs (day laborers, a place to stay, permission to visit and excavate caves, and guides to lead him and his team to said

caves), and what he could offer the villagers in return for assisting him (wages, wells, and the possibility of managing a tourist site).

The archaeological research that followed was largely embedded in village life. We either lived in a communal building in the village (typically classrooms or church kitchens), or in a makeshift camp on the outskirts of town. The archaeological team not only worked closely with village residents while on-site, since the latter made up the excavation team, guides, and survey assistants, but also ate with them, commuted with them, and socialized with them before and after work. Communication in the camp was predominately in Spanish with smatterings of English and Q'eqchi', and as the villagers and visitors became closer, they would find multiple ways to help each other out informally. The archaeologists would provide transportation to town when needed and hire and house masons to improve village wells and other infrastructure when possible. The villagers would repair damaged boots and tools, cook local delicacies, and let the team know about new sites and finds.

The breaking down of barriers between archaeologist and community member soon became the cave subproject's default approach due to its relative ease of working in a heavily populated yet still marginalized corner of Guatemala. In 2003, Woodfill was invited to join a government initiative to convert the Candelaria Caves (Figure 3), the second-largest cave system in Central America, into a national park. He joined a team composed of leaders of three villages, a local non-governmental organization, several applied anthropologists and Peace Corps volunteers, and specialists from the Guatemalan government, and was tasked with documenting archaeological zones within the caves, suggesting low-impact tourist paths through parts of the system, and training the villagers in proper caving techniques to slow down the destruction of the fragile subterranean environment. In return, he received funding from USAID, unrestricted access to the caves under community management throughout the system, a rotating team of guides and excavators, and a place to stay in each village. The park was officially inaugurated in 2004, and Woodfill continued to work in the system until finishing his dissertation fieldwork in 2006.

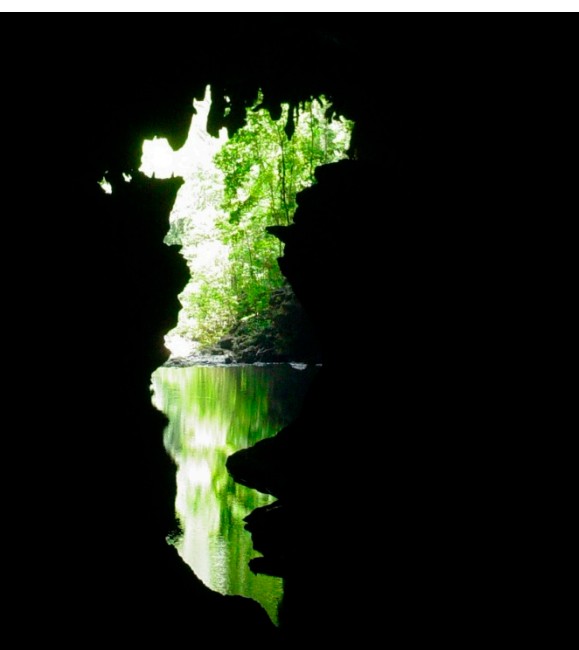

**Figure 3.** Riverine entrance to one of the Candelaria Caves. Photo by B.K.S. Woodfill.

*2.4. Proyecto Salinas de los Nueve Cerros*

In 2009, Woodfill and his colleague Jon Spenard returned to Guatemala to scout out potential archaeological projects when a happenstance meeting occurred between the latter and a group of Peace Corps volunteers who were looking for an archaeologist to work with one of their colleagues.

Volunteer Ted Joseph was stationed a few hours west of the Candelaria Caves system and had spent his term developing a unique forested environment—a massive salt dome, brine stream, and salt flats at the base of the Guatemalan highlands—for community-run ecotourism. In addition to its natural bounty, the salt source was located in the center of a major archaeological site, the massive city of Salinas de los Nueve Cerros (Figure 4), which had been the focus of several small-scale investigations beginning in the 1970s [41,44,46]. From the archaeological perspective, a new project there would answer questions about the Classic Maya economy and the degree to which the Maya elite were involved in the production of basic commodities like salt. The community was excited about the possibility of including an archaeological component to the ecotourism project to strengthen the draw for tourism.

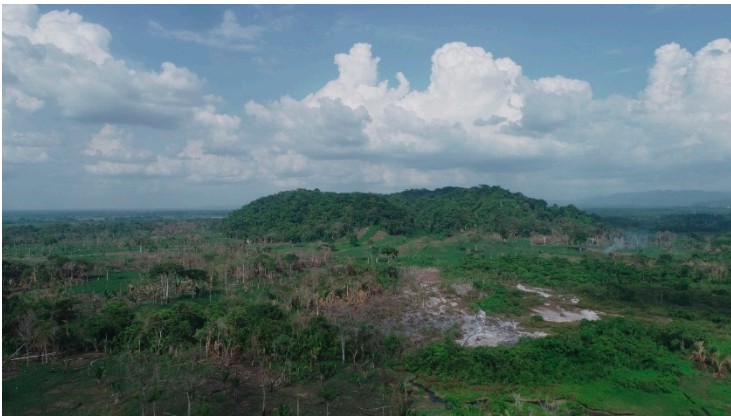

**Figure 4.** Aerial photo of the salt dome and salt flats in the heart of Salinas de los Nueve Cerros. Photo by B.K.S. Woodfill.

The archaeologists traveled to the region in August and met with Joseph and local leaders to tour the site, observe potential research loci, and discuss the potential for collaboration. We were surprised to hear that the impetus for inviting the Peace Corps volunteer into the region to help create the community ecotourism initiative was a visit to the Candelaria Caves by local youths; they were just as surprised to know that we played a part in the park's formation. By the end of the year, Proyecto Salinas de los Nueve Cerros was formed with a team composed of Guatemalan, American, and French students and professionals focusing in archaeology and ethnography, with funding from the Alphawood Foundation and permission from both the local communities and the Guatemalan government.

We assembled a team of excavators representing the four villages most open to working with us. Investigations were to focus on the site core, in land that was owned by the municipality of the distant city of Coban, who shared co-management with a local village organization. As some of the last forested land in the region, this municipal property was the focus of the ecotourism project and housed the saltworks, a brine stream, salt flats, and multiple neighborhoods with monumental architecture. We believed that we could also use research in this neutral territory to acclimatize the Maya horticulturalists who owned small plots that covered the rest of the ancient city to our research methods, allowing us, we hoped, to move gradually into community land in subsequent years. After receiving the approval of both the village organization and the municipality, we headed into the field in March, 2010 to begin our inaugural season.

In spite of (or, more accurately, because of) the strong community support, the project hit a major snag on its first day in the field. Woodfill has discussed this in more detail in other publications [9,24,50], but fundamentally, we ran across two major political problems. The mayor who had recently taken office belonged to a conservative political party that was openly antagonistic towards the communities surrounding Salinas de los Nueve Cerros, and both this mayor and the new municipal council were more interested in petroleum exploitation than community development. Two hours into our first day of fieldwork—as we were setting up test units—municipal workers arrived to evict us, having changed

their mind about granting us permission. So, even after we obtained written federal authorization and verbal support from the communities and the municipality, the project shut down just as it began.

Although the situation seemed hopeless, the workmen organized themselves over the next two days to offer up their own parcels to archaeological investigation. We spent the rest of the field season mapping and excavating mounds underneath village cornfields (Figure 5) with full community support while we attempted to negotiate re-entry into municipal lands. The following season, municipal relations improved, allowing us to live in and fix up an old oil camp in their land (Figure 6) while we continued to work in community parcels. We began the season with a *mayejak*, a ritual offering to the earth spirits, ancestors, and other powerful beings led by local religious leaders that was attended by over 1000 people from throughout the region and served over 250 kg. of chicken (Figure 7, [9]).

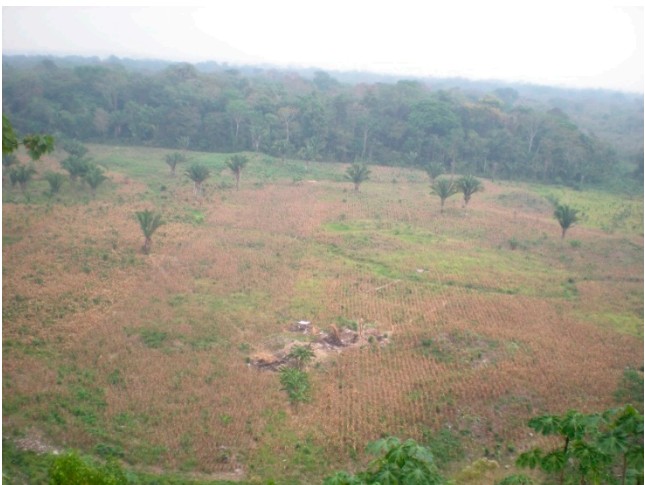

**Figure 5.** Excavations in a horticultural field during the Proyecto Salinas de los Nueve Cerros inaugural field season in 2010. Photo by B.K.S. Woodfill.

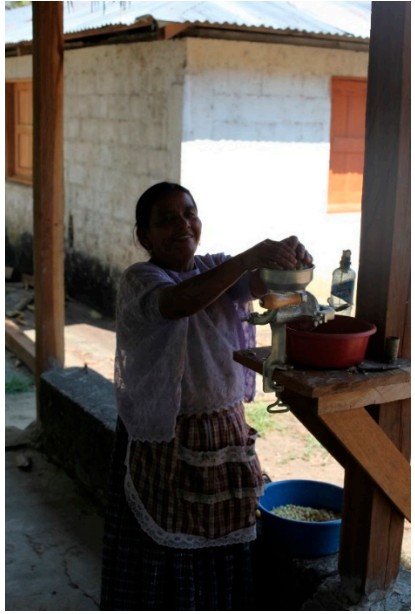

**Figure 6.** Project cook in the newly built field kitchen. Photo by B.K.S. Woodfill.

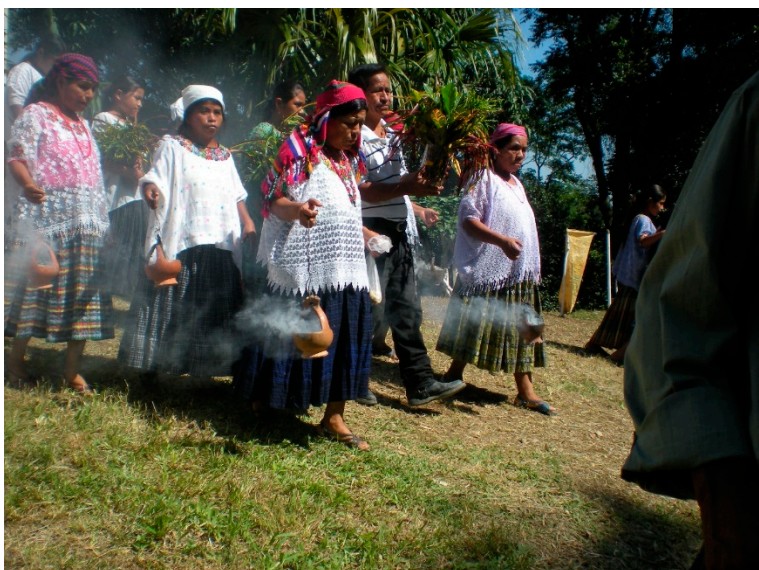

**Figure 7.** Q'eqchi' ceremonial procession to commemorate the beginning of the 2010 field season. Photo by B.K.S. Woodfill.

Although we were initially told that a letter granting permission to work in municipal lands was imminent, that letter never arrived. Instead, we received word half-way though the 2011 field season that council members had sent a team to evict us without warning from the oil camp, and that they were three hours away. Within two hours, the communities had organized themselves again and written their own letter signed by several dozen leaders and landowners protesting this imminent eviction. After the council received news of the protest, they called back their emissaries, allowing us to finish the field season in peace.

The mayor continued to refuse entry throughout his term to not only the archaeological project but to an ongoing long-term biological research project run by professors at Guatemala's national university. The communities fared no better. The previous mayor had secured funding to build a high school on the municipal property before he left office and began construction on it, but after the new administration took over, the rest of the money vanished. After our field season ended in 2011, the co-management agreement with local leaders relapsed. The ecotourism project was officially terminated, all municipal development projects were canceled, and the villagers were now prohibited from entering any part of the property that they were not leasing for farmland.

Historically, the lack of community investment in and access to the municipal land paved the way for invasions from landless poor looking for a place to settle. Such an event occurred two separate times in the 1990s, inspiring the co-management strategy taken by a string of mayors before the election of the then-current mayor. We were worried that history would repeat itself after the municipality closed the land to the locals, and it did in 2015. Over 100 families invaded the municipal land in February and divided it between two new villages. Most of the remaining forest was cut down, the municipal guards were forcibly removed, and, even after the mayor was replaced later that year by another with strong ties to the surrounding region, the political fallout and uncertainties from the invasion prohibited us from conducting anything more than a handful of unmanned aerial photogrammetry missions there.

At the time of writing this article, the antagonistic mayor has returned to office, inheriting a municipal property that still hosting two officially unrecognized Maya villages and a foreign oil company. One of the villages made national headlines when they partnered with a drug cartel, transforming a straight stretch of road into a landing strip [51]. This series of unfortunate events drastically transformed the collaboration between the archaeologists and the Maya communities, forcing both sides to rethink their ambitions and expectations.

For most of the history of our research, the archaeological team (1) focused its investigations on community land in other parts of the site, (2) teamed up with some of the Guatemalan biologists who were similarly shut out of municipal land, focusing on reconstructing the paleoenvironment, and (3) worked directly with local leaders to empower them to conduct their own development. The villagers, meanwhile, (1) focused on smaller-scale development initiatives, especially related to water, maintaining infrastructure, and education; (2) teamed up with multiple institutions and ranchers working in the area to create a more diffuse support network; and (3) worked directly with the archaeological project to provide us with field sites.

The resultant collaboration between the scientists and the communities has resulted in multiple benefits to both sides—a feat only made possible by the extreme levels of flexibility and goodwill each stakeholder exhibited. These results are summarized in Table 1; of these, it is worth highlighting a few. The scientific team, which is composed of archaeologists, ethnographers, ethnohistorians, paleoecologists, biologists, and art historians, has produced an average of just under 14 scholarly products per year for the 11 years the project has been in operation. This list includes one single-authored book; 38 published articles, book chapters, and conference proceedings; six student dissertations and theses; eight successful grant applications; and nearly 100 professional presentations.

**Table 1.** Summary of the principal academic and community benefits from the collaborative research undertaken at Salinas de los Nueve Cerros, 2010–2021.

| Collaborative Research Benefits for | |
|---|---|
| **Scientific Team** | **Local Communities** |
| Professional publications: <br>• 1 published book <br>• 10 published articles <br>• 3 published book chapters <br>• 26 published conference proceedings <br>• 9 publications currently under review or in press | Infrastructure improvement: <br>• 60 wells <br>• 18 bridges <br>• 7 latrines <br>• 2 molds for cement tubes and cinderblocks |
| Student advancement: <br>• 4 dissertation projects <br>• 1 master's thesis project <br>• 6 undergraduate thesis projects <br>• 14+ undergraduate practicums | Sustainable income projects: <br>• 20 women trained in handicraft production <br>• 70+ women involved in managing family gardens <br>• 300+ microcredits to horticulturalists |
| 91 professional presentations at conferences in <br>• The United States <br>• Guatemala <br>• Mexico <br>• Canada <br>• Spain <br>• El Salvador | Land rights: <br>• 87 families resettled with land titles <br>• 200+ families in the process of resettling with land titles <br>• 125 land parcels surveyed for lotification |
| 8 successful grant applications <br>• Alphawood Foundation <br>• National Science Foundation <br>• MACHI/InHerit Passed to Present <br>• National Geographic Young Explorers' Grant <br>• Internal university student grants | Health initiatives: <br>• 15 Ecofiltro water filters donated to schools <br>• 135 Ecofiltro water filters sold at cost to families <br>• 30 discounted low-smoke stoves <br>• 120+ donated eye care and eye glasses |

On the community side, this collaboration has resulted in at least 13 major development projects atop and around the Nueve Cerros archaeological site. These include infrastructure development, sustainability initiatives, public health, clean water, and land rights programs. Over 100 families in at least 15 villages have benefitted from the cooperation between the archaeologists, NGOs, local churches, and community leaders.

While many of the results reflect the differing values and goals of each stakeholder, it is worth pointing out that there are several products that show where a common interest in heritage, the Maya

past, and the potentialities of archaeology have come together. The project provided a springboard for two local Q'eqchi' individuals and a third from the city of Coban who were interested in studying archaeology and anthropology. These three individuals were provided free housing in Guatemala City, a paid position on the project with flexible hours, and practicum and publication opportunities while they attended the national Universidad de San Carlos. They have since co-authored six publications in the Guatemalan archaeology symposium and eight presentations to date, and the first of the three is currently in the process of writing his undergraduate thesis.

### 2.5. Proyecto Sak Balam

The ongoing political problems at Salinas de los Nueve Cerros pushed us out of the salt production zone and into the surrounding region, and in 2017, Woodfill founded a second archaeological project just north of the Guatemalan border in southern Mexico. During the first season, a small team composed of both present authors, co-director Socorro Jiménez (Universidad Autónoma de Yucatán, Mérida), and two Mexican undergraduate students surveyed the surrounding region in search of archaeological sites and caves, largely through organizing community meetings and interviewing village representatives. Outside of a small secondary site in the Nueve Cerros hinterlands, however, there was very little archaeological potential on the Mexican side of the border, although we did hear some promising leads at an established village ecotourism project 30 km to the northwest.

AraMacao Las Guacamayas (Figure 8) is a community-run hotel complex in the Chinanteco village of Reforma Agraria. Located on the banks of the Lacantún River, it is adjacent to the largest nature reserve in Mexico, the 331,200-hectare Montes Azules. During a brief visit that year, we heard about several archaeological sites that local tour guides had visited within the park, so Woodfill returned with Mexican archaeologist Ramón Folsch the following year to continue the archaeological survey. They visited four archaeological sites during the 2018 field season—two small ancient villages in the immediate vicinity of the village; El Palma, a major site that was registered in 1976; and an unregistered site with a Late Classic hieroglyphic staircase in the village of San Isidro.

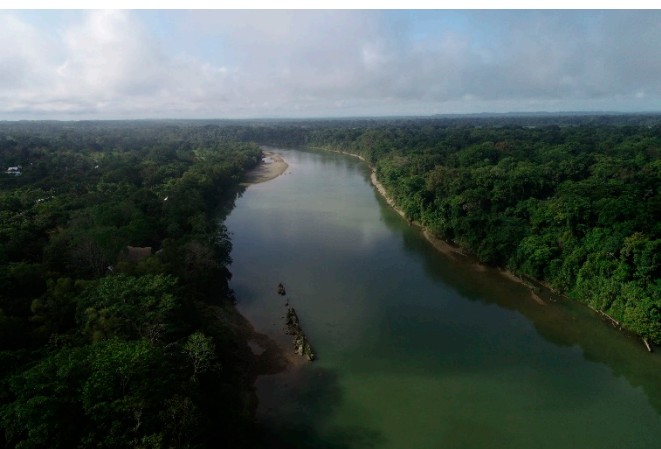

**Figure 8.** Aerial photo of the Ara Guacamayas ecotourism center. Photo by B.K.S. Woodfill.

Woodfill and Jiménez returned with a larger team in 2019 with two goals: fully documenting and conducting a preliminary investigation at San Isidro and El Palma, and kayaking to the headwaters of the Tzendales River to find two known archaeological sites that have never been fully documented, Late Classic Tzendales and Colonial Sak Balam, the last of the Ch'ol kingdoms to be conquered by the Spaniards in 1695 [52]. We accomplished the first goal and made great strides in the second, and, while we planned to continue this line of inquiry in 2020, the COVID-19 pandemic has pushed our follow-up field season back to 2021.

Our research in Mexico, while affected by different problems, histories, and potentialities, continues to be rooted in the same broad philosophy of partnership to address the concerns of different stakeholders. The leaders of Reforma Agraria are interested in expanding the ecotouristic experiences they offer, so our research is directly relevant to their goals. Similarly, the residents of San Isidro were working with a local schoolteacher to develop the archaeological site, located within a village nature preserve, for community-led tourism until the little funding they had secured dried up. We are now in the process of solidifying the collaboration among the archaeological team and the two villages and plan to focus most of our research energy here in the coming decade. At present, though, this project is still in its early stages and is, therefore, still based more on good will and interest than actual accomplishments.

## 3. What Can Happen

As the above discussion illustrates, community archaeology is full of pratfalls, opportunities, challenges, and surprises. Both the community and the academic partners come into the collaboration with specific needs and goals, but as the project develops and the political context evolves, we have to reassess and renegotiate our obligations and expectations as outsider scholars in relation to the communities.

### 3.1. The Bad

Although archaeological practice is typically thought of as fairly straightforward—apply for a permit to excavate a site, set up camp, conduct field research, bring materials back to a permanent laboratory, and write up the findings—the reality is rarely so simple. In addition to more universal problems like inclement weather, global pandemics, slow-moving bureaucracies, and the availability of funds, the three archaeological projects described above are located in a region known for its political and economic precarity [34–36]. This has manifested itself on several levels, including:

1. the appearance of clandestine landing strips associated with international drug cartels who are suspicious of our intentions and research parameters;
2. the power struggle between the mayor of Coban and the local communities;
3. the invasion of the Nueve Cerros site core by landless poor;
4. the proliferation of African palm plantations, oil extraction, and hydroelectric dams leading to site destruction and flooding;
5. the need to negotiate with each landowner individually or in small blocks for access to parts of archaeological sites;
6. turnover in land ownership even after successful negotiations, as owners fall into debt, gather resources to emigrate to more developed regions, or otherwise sell or bequeath their plots.

All of these uncertainties require the investigative team to be exceedingly flexible, necessitating quick shifts in research focus and design, site choice, and personnel decisions. As project director, Woodfill tries to chip away at the site and gather as much data as possible while shifting focus to different neighborhoods or even moving out into the hinterlands and across borders as availability permits.

This is not a luxury as easily affordable to students, potentially requiring them to exhibit even more short-term flexibility and quick thinking to finish their investigations. While conducting dissertation research between 2001 and 2006, Woodfill shifted focus three times to different cave systems due to a combination of restricting access and new opportunities. When actually writing his dissertation (60), he was able to construct an overarching theme—using changes in ritual practice at the different shrines, each of which was on or near the same trade route—to construct a model for understanding how trade was organized.

Rivas's dissertation research at Nueve Cerros similarly required shifting focus from investigating an early ceremonial group to a series of residential mounds and depressions in order to understand ancient water management systems [53]. This change in the research plan was initially difficult to prepare, but became crucial for understanding the Nueve Cerros aquifer, hydraulic systems,

and mound-building practices. Additionally, these investigations were appropriate for understanding ancient landscape practices of a Nueve Cerros neighborhood in relation to sustainability and resilience.

The collaborative model is also potentially problematic for communities. We have observed first-hand multiple issues and have heard local collaborators express frustration at others that had not occurred to us. This list includes:

1.  excavations in, and increased traffic through, cornfields can damage crops, and
2.  the presence of foreigners can pique the interest of neighbors, leading to more looting and traffic.
3.  Since the presence of the archaeological team is contingent on continued funding, interest, and the rhythms of academic life, the arrival, departure, duration, and intensity of the scientific investigation can vary from year to year and will inevitably end.

It is fundamentally impossible to fairly distribute the immediate benefits of our presence in an area. The villages surrounding the Nueve Cerros region are home to over 15,000 residents, and, even in a good year, we only employ about 70 people as cooks, clothes washers, and field hands. Although the local NGO includes members from villages that do not own any land atop the archaeological site, the people we know best and have supported most strongly are those we know, typically because they own land of interest to us or are related to those who do. Only after we were in the region for eight years, for example, did we learn that most of the development benefits were limited to property owners and their families, even though there is a substantial homeless population in the region.

This realization has led us to ask several questions. Is it possible for archaeologists use their presence to benefit families and communities that do not have archaeological remains within their parcels? Can this be attempted for long-term collaboration as well? By focusing more generally on regional development as discussed in the following section, we hope to answer these questions in the affirmative.

### 3.2. The Good

In spite of the negatives discussed in the previous section in mind, there are multiple benefits to engaging in a collaborative research methodology. From an archaeological perspective, these sites would be fundamentally off-limits to a project focused only on Western science and data extraction. Even with funding and legal standing from the state, the reality of post-civil war Guatemala and post-Zapatista uprising Chiapas is that neighboring communities can set up roadblocks, forcibly remove people they declare trespassers, take prisoners, and threaten—and even follow through with—violence. Such potentially drastic measures are sensible reactions to the sustained history of disenfranchisement by the state and outside corporations and investors, which include the Guatemalan genocide, the privatization of the Mexican *ejidos*, and the myriad agricultural, mining, petroleum, and hydroelectric interests that threaten the region.

When collaborative endeavors are well-thought-out and communicated, in contrast, both sides do have something to gain. Local investment in scientific research not only makes research in community lands possible but, in the best of circumstances, encourages community members to share knowledge of potential research sites and unreported discoveries. When the field work is in session, the team is more likely to be protected from potentially negative situations, from the attempted forced removal from municipal lands discussed above to roadblocks, violence, and sequestering.

At the same time, community members can take an active part in the narratives surrounding their region and exert their agency in the types of economic activities and political interventions that occur there. By working closely with national and foreign entities, they can cultivate potential long-term allies with enough international reach to counterbalance some of the negative effects of globalization and modernization that are taking hold in their surroundings.

These research projects can also have a direct benefit through seeking finances. In addition to opening up new lines of grant money provided by development agencies, it can also strengthen applications to more traditional funding sources like the National Science Foundation. Research conducted in collaboration with

descendant communities ties directly into the investigation's "broader impacts". These projects explicitly target the foundation's parameters for assessing such impacts, with "the potential to benefit society and contribute to the achievement of specific, desired societal outcomes" [54].

In order to provide examples of how collaborative archaeology can be beneficial to both descendant communities and scientists in Guatemala, we will pick two examples, each from one of the authors' dissertation projects. The first involves the ongoing relationship between Woodfill and the residents of the village of Mucbilha, located in the center of the Candelaria Caves National Park, and the second involves the cluster of villages on the southern bank of the Chixoy/Salinas River where Rivas focuses his research.

### 3.3. The Candelaria Caves National Park

As the second largest cave system in Central America and the wealth of archaeological material that was already known to be present in the cave system after the pioneering work of Patricia Carot (7, 8), the Candelaria Cave system was ripe for a more comprehensive research project when Woodfill first arrived in the region in 2000. Although he was interested in directing research there, access proved unfeasible for the first few years due to a long-standing conflict between a Q'eqchi' Maya village and a foreign tour operator, which began during the Guatemalan civil war and continued to escalate after the signing of the Peace Accords in 1996. This conflict is described more fully elsewhere [26,36] but will be briefly summarized here.

The village of Mucbilha was founded in 1970 by Q'eqchi' Maya families fleeing the harsh conditions in highland coffee plantations. The swath of land at the base of the highlands that includes the village land was originally set up to be distributed to the landless poor in the 1950s, although after the CIA-backed coup in 1954, much of the land instead was converted to *fincas* (large ranches or plantations) owned by high-ranking military officers and other powerful individuals. Although the first few years of life in Mucbilha was relatively tranquil, trouble from the outside world crept into the villagers' lives with the arrival of a French spelunker who first visited the cave system in 1974. Over the following years, he built up an ecotourism industry with the support of military and political contacts, some of whom owned fincas nearby.

As the civil war heated up in the late 1970s and 1980s, the military was an ever-present threat, and soldiers intimidated and, on multiple occasions, disappeared villager leaders. Although the villagers tried to register their land with the Guatemalan government three times beginning in 1982, each attempt was stalled. After the signing of the Guatemalan Peace Accords in 1996, things were looking up for the community until the foreigner successfully lobbied the government to declare the cave system national patrimony in 1999, prohibiting any land titling within the designated borders. This was followed by pressure to create the Candelaria Caves National Park, which would likely have resulted in the forced removal of the communities contained within it.

The move to declare the national park reached enough momentum in 2002 that preparations began at the federal level, and the Guatemalan government invited the United States Agency for International Development (USAID) to spearhead the effort. Their point person, Anthony Stocks (Idaho State University), was an applied anthropologist who assessed the nature and root of the land conflict and convinced the government to create a new model for national parks for Guatemala, one that involved a system of co-management between Indigenous communities and the Ministry of Culture and Sports. Woodfill was called in to work in village land to assist with the process by registering archaeological remains within the cave system and assisting with creating the tourism infrastructure.

He first met with leaders from Mucbilha and the neighboring village of Candelaria Campo Santo in early 2003 and moved into the villages soon after, spending about eight months over the next two years working in the cave system in tandem with community members. The archaeologists and the villagers each had specific skill sets, connections, and knowledge that they brought to the table and by working together they were able to receive multiple desired outcomes.

The archaeologists were specialists in cave survey, mapping, and the acquisition and interpretation of archaeological remains. As the team included three cavers who were also studying ecotourism at a private university in Guatemala City, we also had knowledge of best practices for creating low-impact tourist trails in the fragile cave ecosystem. We also brought with us access to funding, press, and other specialists. The villagers, in turn, knew the location of many of the caves in the area and those most promising for archaeological investigation. They provided us with a camp site and organized to allow us to hire guides, excavators, cooks, and laundry service. Finally, their participation granted us permission to pass through farmers' fields, although once our association with the village faction was known we became *personas non gratas* in the foreigner's hotel complex and the few caves in land he claimed control over.

The collaboration proved beneficial for both sides as well. The archaeological project resulted in one doctoral dissertation [55], three undergraduate theses [56–58], two books [26,29], six articles and book chapters [59–64], seven proceedings from the Guatemalan archaeology symposium [65–71], and myriad presentations. On the village side, the work was a necessary step for the creation of the park and resulted in the establishment of low-impact tourist paths in six caves. Woodfill wrote a manual for tour guides and arranged an interchange for residents of Mucbilha and other villages engaged in cave tourism to visit established tourist caves in Belize (Figure 9); Belizean tour guides later visited Mucbilha to continue training their local counterparts. After the project investigation was concluded, several national and international funding agencies constructed a small ecotourism hotel and visitor center, a gravel road to this ecotourism complex, stone paths to the caves, and publicity for potential visitors. Most importantly, the Guatemalan government signed an agreement with residents of Mucbilha, Candelaria Campo Santo, and a third village adjacent to the park that allowed their residents to purchase their land at a greatly reduced rate in exchange for protecting the caves and forest within the park system.

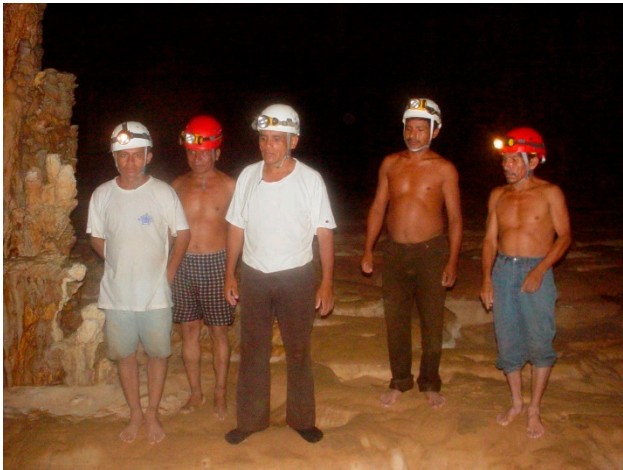

**Figure 9.** Q'eqchi' villagers in Actun Tunichil Muknal Cave, Belize. Photo by B.K.S. Woodfill.

Although the official collaboration between Woodfill and the community ceased in 2004, the relationship has continued and each of the stakeholders continues to assist the other in various matters. The villagers continue to keep Woodfill appraised of new archaeological finds, including a rare pecked cross (Figure 10) within the cave system that was originally reported by Patricia Carot and rediscovered in 2012. The resulting article [60] was largely inspired by a conversation held during a visit to the pecked cross among archaeologists, cavers, and Q'eqchi' leaders, landowners, park guides, elders, and spiritual leaders about what the feature meant to each group.

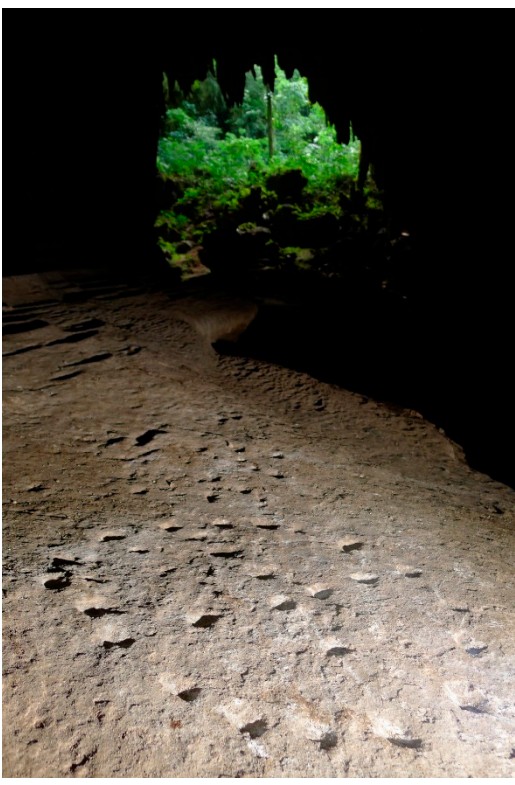

**Figure 10.** Pecked cross in the Candelaria cave system. Photo by M. Oliphant.

Woodfill has also continued to be involved in the local struggle for autonomy. The conflicts with the foreign spelunker and hotelier continued through attacks on the villagers in the press, on social media, and in 2012, in another round of physical violence directed towards the villagers. After this last injustice, the residents of Mucbilha peacefully shut down his old ecohotel, although they were soon accused in the media of using violence themselves. Woodfill worked with the Guatemalan Ministry of Culture and Sports to assess the actual damage the hotel's closure and a recent municipal road project might have caused in the wake of this scandal. Although it was negligible to non-existent, all of the landowners from Mucbilha—over 50 men—were ultimately charged with multiple crimes by the foreigner and his business partner, and a warrant was made for their arrest.

As a result, the men from the village were unable to seek work outside of their village, and several of them fled to the United States. Woodfill and two other Americans who worked in the village, a professor of Geography and a former Peace Corps volunteer, spent six months between 2018 and 2019 helping one of these villagers navigate the evolving judicial landscape for refugees to be granted a legal status for him and his son. After acquiring the *pro bono* support of a major New York law firm; a Texas-based refugee organization; a Bethesda, MD-area church; and a couple from this church willing to host the father and son, the pair were eventually released and are still awaiting a hearing. To help with the Guatemalan end of things, Woodfill, the geographer, the former Peace Corps volunteer, and two anthropologists are each contributing chapters to a Spanish-language edited volume [72] to be published in Guatemala about the village and its history in order to assist with the fight against the human rights abuses its residents continue to receive.

### 3.4. The Nueve Cerros Water Management Project

Rivas began archaeological work in Guatemala in July of 2015. His research interests lie in ancient Maya landscapes, and decided to join the Nueve Cerros team to conduct on-the-ground surveys, locating mounds, water management features (canals, ditches, reservoirs). He arrived in Guatemala 10 days after finishing a field season in a highly technical, Western science-focused archaeological

project in a remote region of Central Asia. The field research teams alone consisted of geoarchaeologists, geophysical remote sensing researchers, an aerial remote sensing specialist, a paleoethnobotanist, a denrochronologist, a geochemist, and an excavation team.

Rivas did not spend the first week in the field in Guatemala conducting archaeological investigations; he instead was sent to survey the corners of every horticultural plot of one of the Q'eqchi' communities built atop the Nueve Cerros site. The farmers needed to know the size of their parcels and other general spatial information to divide them up and begin the process of purchasing the land.

As a first year PhD student, he was open and willing to help the community with their mapping efforts at Woodfill's direction, although he did not understand why an archaeology project was so involved with community development. Rivas does not work for an NGO, is not a civil engineer, nor does he work for the local government that would be assumed responsible for parceling farmers' lots. Up to this point, he had never spent field time conducting anything other than archaeological work. Having an archaeology student spend valuable field time, money, and equipment on anything other than archaeology was unheard of to him, and not something that was prioritized in the four previous heavily funded projects he has worked on.

Rivas has since learned that these surveys are not usually done for the communities until election time, in which local officials running for office release funding for most of the civil engineering projects. He also observed that subsequent interactions with the Q'eqchi' were radically different than he was used to. For the rest of the season, local farmers volunteered their efforts in assisting Rivas in his own landscape interests. When he began his archaeological investigations the following week, he worked alongside one of the local farmers who accompanied him on his survey. The farmer walked with him around the site, making sure other landowners were aware of his presence, and that he was not intruding. Although this was not always a harmonious situation, locals around Nueve Cerros knew that we were not simply scientists extracting information for our own benefit. The surveying efforts were rather successful over subsequent years, and after he finished his doctoral coursework, Rivas decided to focus his dissertation proposal on the E-Group Ceremonial Center (Figure 11) which was slated for excavation during the 2018 field season.

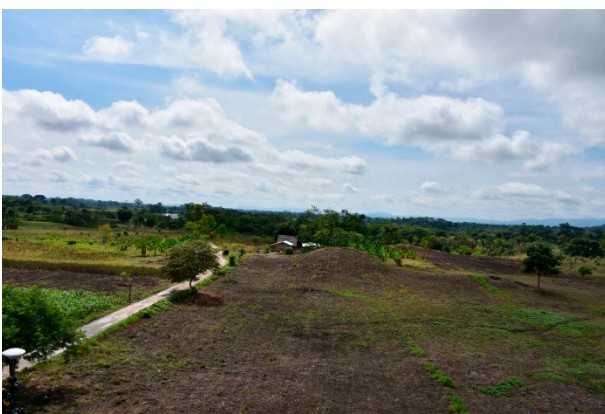

**Figure 11.** The Nueve Cerros E-Group. Photo by A.E. Rivas.

There are clear anthropological reasons for why this center is important to investigate. E-groups can represent early sedentary communities, have astronomical significance, and reveal interregional interactions [73]. The E-group ceremonial center at Nueve Cerros is located between two lots owned by different families. A few months before the field season began, Rivas contacted the owners of both lots, requesting permissions for excavations. He turned in a dissertation proposal and plan for investigations and was ready for carrying out his dissertation project. However, a month before the field season, Rivas had learned from local collaborators that one of the families had sold their lot to an unknown rancher who did not live in the region, meaning we had lost access to part of the ceremonial

center. The other family, which had never directly collaborated with field archaeologists, revoked their permission to excavate their half of the architectural group. The head of the family had recently moved to Florida, U.S, and understandably felt uncomfortable having foreign researchers investigate their land without him being present. Within a week, however, he was able to work with community leaders to propose and obtain permission for an alternate project and resubmitted his dissertation paperwork.

Rivas's actual dissertation focused back on large-scale surveys and residential water management at Tierra Blanca, a sector of the Nueve Cerros site that contains most of the households and residential groups. The investigation shifted from focusing on early ceremonial centers and ritual practices to questions relating to hydrology and landscape construction. Rivas and Woodfill conducted drone photogrammetry and digital elevation models (DEMs) of the major architectural groups at the site including the epicenter, the salt production zone, and the Tierra Blanca area (Figure 12). Rivas then used this data to create hydrological modelling using geospatial analyses tools. Additionally, Rivas excavated residential mounds and depressions at Tierra Blanca. His excavation and GIS modeling results indicate usage of a collective economy and residentially controlled water system based on groundwater wells, dating at least to the Early Classic. The easy access to the aquifer is not common among major Maya cities, which relied on constructed rain-fed reservoir systems. Access to the water table may have played a large role in sustaining the population for survival at least four centuries after the Classic Maya collapse [26].

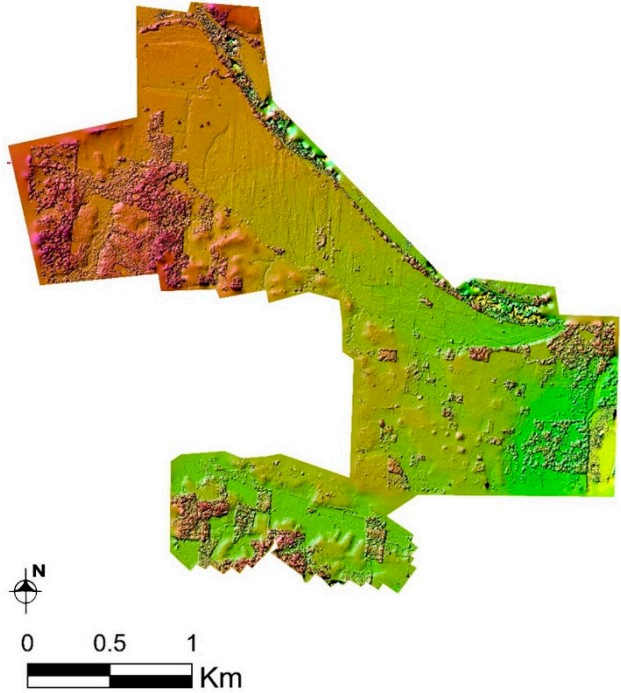

**Figure 12.** Digital Elevation Model (DEM) of the Tierra Blanca sector of Salinas de los Nueve Cerros. Image by A.E. Rivas.

Groundwater access was not just valuable for ancient Nueve Cerros laborers. Today Q'eqchi' families rely heavily on groundwater wells, and the construction of community wells is now one of the most important development projects that NGO's are involved with in the area. The Nueve Cerros project works with ADAWA (Asociación Civil No Lucrativa para el Desarrollo Aj Waklesinel), a local indigenous operated NGO focused on community development initiatives [74]. Public well building is one of the major projects ADAWA is continuously working on, from building large community wells in each of the local schools to providing discounted water filters to schools and individual families. The data Rivas collected on the ancient hydrological system was mutually beneficial for the farmers and archaeologists. As Rivas has described elsewhere [75], farmers living in the Tierra Blanca area

reopened some of these ancient depressions as modern wells and even created a canal system to separate clean and dirty water (Figure 13). This study showed how an archaeological understanding of subtle topographic changes through GIS modelling, and stratigraphy of depressions can directly contribute to community development projects.

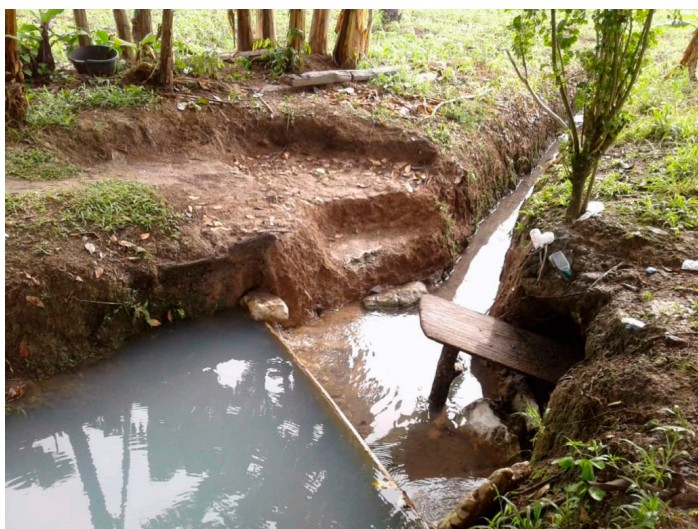

**Figure 13.** Clean and dirty water. Photo by W.G.B. Odum.

His research efforts went beyond the excavation area and recording ancient architecture, with drone photogrammetry of the banks of the Chixoy River, and the hilly terrain west of the Nueve Cerros ridge. This was also done with members of ADAWA, who helped Rivas gain access to different plots of land. In these drone flights, no archaeological mounds were discovered, although some surface artifacts were visible during ground survey. These data were used for understanding the physiography of the greater Nueve Cerros region.

While conducting these surveys, community members living on the banks of the river petitioned Rivas and the ADAWA members for a water filter for the local school. In this particular community, none of the families had filters in their homes, and neither did the local elementary school. Local elders were particularly concerned for the children, as they are well aware of the impact of waterborne illnesses on the young. The elders know about the work ADAWA conducts for the nearby communities, and they seized the opportunity to speak with the NGO members. The following day, Rivas completed his drone work along the river, and ADAWA was able to donate one of their water filters to the local school. The team spent an afternoon at the school, teaching the schoolteacher and students how to properly use and maintain the filter. These filters, made by the Guatemalan company Ecofiltro, are inexpensive, distributed throughout the region by ADAWA, and can be used for up to two years, making them valuable long-term and easily replaced solution for the many Q'eqchi' families.

At a village west of the Nueve Cerros Ridge, community leaders are in an even more dire situation. When we first arrived to this village, the community leaders were very skeptical of the team's presence. Were we here just to extract resources or data for their own ends? Would any of this information be helpful to the community? When the local farmers realized the kinds of surface hydrology information Rivas was interested in, they instantly opened up to him. This particular village has had many problems with accessing groundwater. A large community well was built at the center of the village, with the financial help of a European operated NGO. Geophysical surveys were conducted by the NGO but did not reveal any information on water table depth. Unfortunately, the well runs completely dry during the dry season, making this feature obsolete for the four to six months of the year when it is most needed.

Rainfall is still common during the dry season, so a large tank was placed on top of the well but is not enough for village-wide needs. After the locals explained this situation to Rivas and ADAWA members, they asked us if the topographic surveys can help them identify areas where they can place a well that can remain with water throughout the year. On this particular day, we were also joined by a civil engineer from Coban, who was interested in assisting the development programs we have been working on. While we have not to date built any new wells or planned water systems with the village, talks are still ongoing with all interested parties to further assess how ADAWA, civil engineers from Coban, and the archaeological team can work together for these collaborative interests.

With these examples of community-oriented archaeology, it is clearly shown that the simple presence of archaeologists can also have some long-term benefits for all parties involved. For this to work, archaeologists must think beyond the rigor of scientific investigations, have flexibility in time management of a field season, and an understanding of the value archaeological research can bring to the community, from heritage programs to local infrastructural needs. These projects have shown that aside from documenting ancient societies and extracting data, community members can tell us directly how they can benefit from the data we are collecting from their homes.

## 4. Discussion

As archaeologists and community members navigate the unexpected challenges and opportunities in a shifting political landscape, there are still several things that all stakeholders should keep in mind to make the collaboration as mutually beneficial as possible. To that end, we echo Smith [12] and others [2,3,7,9,76] in suggesting several characteristics which should increase the odds of success for collaborative research projects.

1. Each Stakeholder Group Should Be Able to Address Needs of Other Stakeholders

On the academic side, needs include a reasonable amount of freedom and support to acquire the data necessary for theses, dissertations, presentations, and publications as well as any letter of support needed for grants and government agencies. The community should provide the necessary infrastructure—campsites, laborers, food, water, etc.—for the field component of the research at a reasonable cost to the researchers as well as permission to enter land, extract materials, and, if necessary, transport each season's finds to a project laboratory.

Community members, in turn, should be able to expect the archaeological team to contribute towards working through their pressing issues and problems through their expertise, equipment, and connections. In our experience, these include land rights, legal battles, the development of sustainable income, access to education, and water management. Some of the archaeological skillset—successful grant-writing, hydrological modeling, survey, soil analysis, and mass communication—can be easily refitted to address needs and problems that locals address. We have worked with issues of land use, water management, sustainable income generation, increasing tourism, and helping with human rights abuses.

2. Each Affected Community Should Support the Presence and Involvement of the Other Stakeholders

Collaborative research affects the entire scientific team to varying degrees. When non-scientists are able to voice concerns and have input in research parameters and design, investigators have to be somewhat flexible. Of course, the idea that archaeologists can actually perform fieldwork without some level of community support is largely a fiction. Even when communities are not explicitly included in planning and executing scientific investigations, there are myriad examples in Guatemala and Mexico when they exert their will through more forceful means, including roadblocks, kidnapping, vandalism, and legal challenges. We feel that it is better to have the flexibility up front, to be explicit about the possibility of changed plans when forming the research team, and to have mechanisms in place to accommodate changes in research venues and timelines as they come up.

In order to acquire community support, we have found in general that the best way to open a dialogue with community members is through a meeting with designated village representatives—the Community Development Committee (COCODE) in Guatemala and the commissar (*comisario ejidal*) in Mexico—in which we clearly state reasonable goals, expectations, and possible avenues for community

assistance. If successful, we set up a time to meet up with the village as a whole (which, in our experience, can occur within a few hours or a few days). The larger meeting has the same goal as the previous one, in which each side is able to delineate what it is able to reasonably offer and expect.

3. Limit Most of the Collaboration and Negotiation to the Specific Individuals and Organizations That Bring Something Concrete to the Table

In our experience, only a small part of the community and the research team is actually involved in any collaborative efforts beyond this initial meeting. The lab technicians and specialists typically spend little time in the field and have little to offer to any development efforts, and since research tends to focus on specific parts of community land, most locals are unaffected by our presence. Different local factions end up being the primary collaborators, depending on the specific political landscape of the community. Over the past twenty years, Woodfill has worked closely with political and religious leaders, ad hoc landowner associations, community non-governmental organizations, schoolteachers, and Christian congregations to acquire the required permissions, support, and knowledge needed to conduct field research. There has been an equal amount of variation on the project side, from trained applied anthropologists and ecotourism specialists to open-minded project directors and field archaeologists who have a sought-after skill set.

## 5. Lessons Learned

As collaborations between academics and communities evolve, flourish, and, occasionally, sputter, there are a few basic characteristics that all sides should agree on to ensure continued success.

1. Each side must be willing to engage in sincere and sustained dialogue, and must accept that no one group will get everything it needs.

In general, archaeologists working in the Maya world do not have the multigenerational weight of mistrust and misdeeds that North Americanist colleagues have been forced to reckon with since the signing of the North American Graves Protection and Repatriation Act in 1990 [77]. However, scientific teams are typically composed of individuals who come from much more privileged backgrounds and situations than the local populations they encounter in the field, which leads to its own problems of mistrust and mismatched values, assumptions, and prejudices from all parties. Each stakeholder group must try to work past them to be able to engage with the others. In addition, each group must be willing to compromise and prioritize goals.

2. Each side must be flexible and have reasonable expectations, both for the total amount of benefits they can expect and the timeline for when these can occur.

This has been discussed in more detail in an earlier article (62), but the biggest sources of conflict and frustration seen by the authors have occurred when one or more stakeholders are unwilling to compromise or be patient. Funding, organization, data analysis, write-ups, permits, and shipping can all take longer than expected, especially because many of them also depend on other bureaucratic institutions—universities, foundations, and government agencies. In the instance when access to land for archaeological work is cut, all aspects of the research project—research questions, timeline, data plan, crew—needs to be flexible, with alternative plans as a realistic option to continue fieldwork. In these cases, long-term relationships and collaborations with communities can be crucial, as having positive relationships with different families, farmers, and local leaders can lead to new opportunities for archaeological investigations.

3. Each stakeholder must take the others into account when making plans that affect them.

On the academic side, we need to abandon the traditional mindset in our field, that our research benefits a nebulous "humanity" that allows us the comfort of being neutral observers who simply report facts. Instead, we need to think about how our research can be beneficial or detrimental to the other stakeholders, fundamentally following McAnany [76], p. 52 by attempting to "rebalance what were unidirectional power relations and augment multivocality". What we publish, where we dig, how we present our finds can all result in distinct and profound outcomes for the people who live on the surrounding land, which can lead to increased looting, foot traffic, elevated land prices,

and possible legal troubles. While we absolutely do not encourage intellectual censorship of any kind, we do feel that it is high time for archaeologists to think through the practical and ethical ramifications of our research to the extent that our colleagues who deal with living human subjects do.

On the community side, landowners and local organizations need to keep investigations in mind when building, sowing, creating infrastructure, etc. The communities must be committed to preserving archaeological remains for the archaeologists and, whenever possible, giving us sufficient warning about pending sales, construction projects, resource extraction, etc. that could damage the archaeology so we can shift emphasis there.

## 6. Conclusions

Archaeologists typically understand our field in one of three ways. For many, especially those couched in the "archaeology as science" camp, our investigations are thought to be politically neutral, based in acquiring evidence of the human past, which is used to more broadly understand the history and diversity of our species. The second and third camps view archaeology as a potential boon for affected communities, albeit in different ways. Proponents of the second camp point to the inherent value of archaeology outside of academia, be it through developing Third World nations and marginalized communities, documenting and promoting evidence of past greatness and pride, or allowing descendant communities to reclaim some degree of control over their heritage. The third camp, of which we are obviously a part, seeks to move beyond the small piece of the Venn diagram where archaeological and Indigenous interests intersect—interpreting and preserving the past. Instead, archaeological methodology and presence can be repurposed as a set of tools that can address a much broader range of issues that are identified by the community members themselves, just as archaeologists depend on community support to investigate the minutia of the human past.

Regardless of the focus, all three of these approaches promote the idea that the field is providing some sort of greater good, be it scientific knowledge, sustainable income, or weapons to fight against racist narratives. While these are certainly beneficial to varying degrees, archaeology still uses methods and theories rooted in the discipline's colonialist history. Archaeology is still fundamentally an extractive discipline that benefits the careers of the scientific community but provides virtually no long-term advantage to the communities surrounding our field sites. When sites are developed for tourism, the new regulations, skyrocketing property values, and education and linguistic barriers for the well-paying jobs that are created often alienate and force out those with whom we worked [26,27,63,78,79]. Furthermore, while it is certainly important for descendant communities to reclaim their past from the long list of problematic and often racist narratives involving aliens, Israelites, Phoenicians, and the inevitability of European world dominance, this is often a much less pressing concern than many other effects of institutionalized racism and marginalization that these communities experience.

Collaborative community archaeology, as discussed throughout this article, is one way to open up our research to include—in a meaningful and mutually beneficial way—the communities already affected by our research. By transforming neighboring communities from pools of labor into active stakeholders, the archaeological endeavor can be used to address interests and concerns beyond our own narrow focus on the human past. By repurposing our presence, skills, and knowledge in ways that can be used to support and amplify local initiatives, we become each other's accomplices while still advancing our own agendas. By reframing archaeology in this light, both Woodfill and Rivas believe that we can truly begin to decolonize our field.

**Author Contributions:** Conceptualization, B.K.S.W. and A.E.R.; methodology, B.K.S.W. and A.E.R.; software, B.K.S.W. and A.E.R.; validation, B.K.S.W. and A.E.R.; formal analysis, B.K.S.W. and A.E.R.; investigation, B.K.S.W. and A.E.R.; resources, B.K.S.W. and A.E.R.; data curation, B.K.S.W. and A.E.R.; writing—original draft preparation, B.K.S.W. and A.E.R.; writing—review and editing, B.K.S.W. and A.E.R.; visualization, B.K.S.W. and A.E.R.; supervision, B.K.S.W. and A.E.R.; project administration, B.K.S.W.; funding acquisition, B.K.S.W. and A.E.R. All authors have read and agreed to the published version of the manuscript.

**Funding:** This research was funded by the Alphawood Foundation, the United States Agency for International Development, Chemonics, Incorporated, Fortalecimiento Institucional de Polítics Ambientales Water Fund,

the Foundation for the Advancement of Mesoamerican Studies, Incorporated, and the Vanderbilt Institute of Mesoamerican Archaeology. Maya Area Cultural Heritage Initiative/InHerit.

**Acknowledgments:** The authors would like to thank Special Issue Editors Chelsea Fischer and Arlen F. Chase for inviting us to contribute to this important endeavor. We would also like to thank the communities of the Ecoregión Lachua and the Candelaria Caves National Park for the decades of comradery and support and the Guatemalan Instituto de Antropología e Historia for research permits. Finally, we would like to thank the editors of *Heritage* and the three anonymous peer reviewers for making this a stronger and smoother article.

**Conflicts of Interest:** The authors declare no conflict of interest.

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
