# Peer review of "Addressing Problems beyond Heritage, Patrimony, and Representation: Reflections on Twenty Years of Community Archaeology in the Southwestern Maya Lowlands"

_heritage, doi:10.3390/heritage3030033_

Round 1
Reviewer 1 Report
Thank you for the opportunity to review this manuscript. The inclusion of descendant communities and other stakeholders in archaeological practice makes what we do more meaningful. With that said, I only have a few suggestions that, I hope, will strengthen the manuscript:
- This is a community engagement/collaborative paper but I would expect that there would be a local collaborator (who is not an archaeologist) as a co-author. I understand that this is a reflection of the authors but the long history of colonialism and the hegemony of Global North archaeologists in the study area make it important that the communities that they work with are represented in the article. If we do not provide that space and relinquish some control to our local collaborators, we are just reprising the colonial and condescending views of western scholarship.
- Related to #1 above, collaborative archaeology also forefront Indigenous epistemology. I would like to see this emphasized in the essay as this does not only provide a space for local views, it is also empowering. If these two items are not included in our work, we cannot call our approach 'engaged," rather, it will only fit the outreach category.
- The essay can also elucidate the role of the archaeologists in the development of heritage narratives in the lowland Maya (Mexico-Guatemala) region. As these communities are swallowed by assimilationist policies of nation-states, how are we, as archaeologists who work directly with Indigenous/local communities, can empower them through the products of our work. The manuscript did mention some (income from tourism), but how about ideas about their history?
- There is no single definition of community archaeology, but there is a commonality in this approach among archaeologists that explicitly involve local collaborators. Perhaps the authors can refer to the Journal of Community Archaeology and Heritage for some guidance: https://www.tandfonline.com/action/showAxaArticles?journalCode=ycah20.
- On the onset, the manuscript explicitly mentions that "archeologists have embraced the inherently political nature of our field and worked to share some tangible advantages with the communities most affected by our presence and interpretations.” I was surprised not find Randy McGuire's work in the bibliography, particularly, Archaeology as a Political Action (UC Press 2008).
- What are the authors advocating in the collaboration? Space for local communities to develop Indigenous archaeology? The most recent JCAH publications referenced above might be useful, esp. the work in the special issue focused on Southeast Asia and Gonzales and Edward's paper.
Author Response
We appreciate the sentiment and points made by this reviewer about the field and its complicity in global geopolitics. We are responding point by point to the comments that came back to us.
- We agree that co-authorship with Indigenous individuals is necessary, and we want to support the inclusion of Maya voices in more spaces. That being said, co-authorship is a large commitment of time and energy, and a specialized, English-language journal, even one that is open-access, would have few direct benefits for Q’eqchi’ Maya we regularly collaborate with. We do co-author with Indigenous archaeology students and local leaders in the Guatemalan archaeology symposium, field reports, grant applications, and other publications and presentations related to their research and development interests.
- Yes. Again, we agree, but we tried to make more explicit our belief that they should be guiding us to ways to help them that are most helpful and meaningful. Epistemology is important, but regular access to clean water, land rights, and basic health care are prioritized more highly than heritage among the communities we collaborate with, so we help them with those and help them find their voice when it comes to grantwriting and government petitions.
- Working to uncover their history and than sharing it with them is important, but we feel it is at least as important to listen to their histories, since archaeology is but another way of knowing embedded in a distinct and idiosyncratic ontology that cannot necessarily be prioritized over their own. The Maya have had 500 years of white people cataloging and typologizing them, so we try to emphasize communication and, again, ways of collaborating that help them work towards their ends instead of prioritizing the struggles that more closely hew to our own interests. We do have an annual ceremony and presentation in the communities at the close of the field season in which we present in Spanish and Q'eqchi' our findings, honor the workers, landowners, and others who made our work possible, and have discussions with elders and spiritual leaders about what the finds mean.
- We have drawn from more sources to explicitly define community archaeology and how we are pushing against the subfield as commonly understood.
- We are not familiar with that book but look forward to reading it when life returns to normal post-COVID. One of us recently defended his dissertation and is in the middle of revisions; the other teaches at a school with a limited digital and physical library, and unfortunately there was not time to get a hold of that book.
- We tried to be more explicit about what we are advocating for both in the manuscript and our earlier comments here. We want to use our presence to empower communities to work towards their own ends. While that can and does include providing space for some of the community members to practice archaeology, archaeologists in Guatemala need not only regular financing but must graduate from college and pay a monthly membership to the humanities guild, while most of the people in the region have not graduated from elementary school. We are supporting regional leaders who are working to provide more access to education (and more income so that families are able to send their children to school to take advantage of the education that is available), but it is simply unfeasible at this point to push the locals to create an Indigenous archaeological program.
Reviewer 2 Report
I congratulate the authors on their paper. I found this article engrossing and the arguments and content were well supported. I appreciated the reflexive and self-critical approach to your work and I applaud your attempts to decolonise your approaches to descendent communities.
I recommend publication with minor changes. I have a few comments and suggestions for small revisions. I suggest that the authors should be free to accept or decline my suggestions as they see fit.
For the editors: there are a couple of formatting issues lines 113-114 (drop image down to next page) and 672 (extra space)
Author Response
Thank you! Unfortunately, we did not receive a copy of the manuscript with your markup, but we did copy edit, clarify, modify, and expand the document to address problems we saw and that were mentioned by the other reviewers.
Reviewer 3 Report
This is an interesting manuscript about research undertaken in the Maya lowlands. I am not an expert in this geographical region however and found that i was in the dark a lot of the time as to the significance of the area, the archaeological context etc.
Overall, the paper is presenting the story of the authors experiences working in the area in a 'collaborative' context.
I have presented in-depth comments in the attached manuscript however i will mention a couple of key points here:
the study must be embedded in the literature around community archaeology, collaborative archaeology, partnership-based archaeology, postcolonial archaeology etc. At present the study barely references any literature from the past two decades to this research approach. However, by doing so, the authors will be better placed to embed their findings into this broader field. Are you conclusions/identification of key characteristics different from others that authors have identified from across the globe? Or in the southwestern Maya lowlands? What exactly is collaborative archaeology as you are using the term? What are you bringing to the discussion that is new, different etc.? I would also like to see a historical backdrop to the region for the international reader.

Author Response
Thank you for your comments. We have tried to integrate our approach more explicitly with community archaeology in general as well as given a bigger introduction to the region, the history, problems, and concerns of the local communities, and through this, hopefully, an explanation for why our community archaeology approach took such a different turn than the norm.
We addressed the comments in your .pdf of the manuscript.
Round 2
Reviewer 3 Report
The authors have made significant changes to the article that explain more clearly the aims and relevance of the project but also to the description of the study area and other matters needed to explain the project to a broader, international audience.
The section on community archaeology is still relatively brief and doesn't delve into how this project adds another dimension to the field but the material on decolonisation, especially with Smith, and Atalay is good to see.
There are a few small matters that i have raised in the attached PDF but they should be easy to deal with.

Author Response
These are great and helpful revisions. I only left "site core" unmodified, since I figure that it is a term used in areas outside of Mesoamerica as well.